# Interleukin-23 levels in umbilical cord blood are associated with neurodevelopmental trajectories in infancy

**Machiko K. Asaka[1], Tomoko Nishimura[1,2], Hitoshi Kuwabara[3,4]\*, Hiroaki Itoh[5], Nagahide Takahashi[1,6], Kenji J. Tsuchiya[1,2]\***

1 United Graduate School of Child Development, Osaka University, Kanazawa University, Hamamatsu University School of Medicine, Chiba University and University of Fukui, Suita, Japan, 2 Research Centre for Child Mental Development, Hamamatsu University School of Medicine, Hamamatsu, Japan, 3 Department of Psychiatry, Hamamatsu University School of Medicine, Hamamatsu, Japan, 4 Department of Psychiatry, Saitama Medical University, Moroyama-chou, Saitama, Japan, 5 Department of Obstetrics and Gynecology, Hamamatsu University School of Medicine, Hamamatsu, Japan, 6 Department of Child and Adolescent Psychiatry, Nagoya University Graduate School of Medicine, Nagoya, Japan

\* tsuchiya@hama-med.ac.jp (KJT); kuwabara@saitama-med.ac.jp (HK)

**Data Availability Statement:** All relevant data are within the manuscript and its Supporting Information.

## Abstract

Our previous study, which aimed to understand the early neurodevelopmental trajectories of children with and without neurodevelopmental disorders, identified five classes of early neurodevelopmental trajectories, categorized as high normal, normal, low normal, delayed, and markedly delayed. This investigation involved measurement using the Mullen Scale of Early Learning in a representative sample of Japanese infants followed up from the age of 0 to 2 years (Nishimura et al., 2016). In the present study, we investigated the potential association between cytokine concentrations in umbilical cord serum with any of the five classes of neurodevelopmental trajectories previously assigned, as follows: high normal (N = 85, 13.0%), normal (N = 322, 49.1%), low normal (N = 137, 20.9%), delayed (N = 87, 13.3%), and markedly delayed (N = 25, 3.8%) in infancy. Decreased interleukin (IL)-23 levels in the cord blood were associated with the markedly delayed class, independent of potential confounders (odds ratio, 0.44; 95%confidence interval: 0.26–0.73). Furthermore, IL-23 levels decreased as the developmental trajectory became more delayed, demonstrating that IL-23 plays an important role in development, and is useful for predicting the developmental trajectory at birth.

## Introduction

Neurodevelopmental disorders (NDDs) are defined as disorders with distinctive features that occur during a child's developmental period. It is common for these disorders to start at an early stage of development, often before starting school. The types and extents of developmental disabilities range from challenges in learning specific skills or executive function control, to those that affect a child's social skills, language, motor function, and intelligence [1]. Often, two or more symptoms categorized as NDDs can be simultaneously observed in a single child.

**Funding:** This work was supported by grants from the Ministry of Education, Culture, Sports, Science and Technology in Japan (https://www.mext.go.jp/en/index.htm, JP20K07941 to TN; JP20H03601 to HK; JP21K07479 to NT; JP19H03582, JP21KK0145, and JP22H00492 to KJT), Japan Agency for Medical Research and Development (https://www.amed.go.jp/en/index.html, JP21gk0110039 and 23gn0110079h0001 to KJT). The funders had no role in the study design, data collection and analysis, decision to publish, or preparation of the manuscript.

**Competing interests:** The authors have declared that no competing interests exist.

For example, children with autism spectrum disorder (ASD) often demonstrate intellectual disability or language disorders, and many with attention-deficit/hyperactivity disorder (ADHD) develop specific learning or tic disorders [1–4]. Findings from family and twin studies have shown that genetic contributions to psychiatric disorders do not align with diagnostic categories in all cases [5]. However, genome-wide association studies have shown that specific single nucleotide polymorphisms (SNPs) are associated with a variety of psychiatric disorders, including NDDs [6].

It is scientifically and clinically important to capture the representations of the biological bases of a group of conditions, also referred to as endophenotypes, that precede the emergence of each condition, in order to move beyond the descriptive categories of NDDs. Early neurodevelopmental trajectories, defined as longitudinal developmental patterns captured by behavioral features such as motor function and language, are examples of endophenotypes, as they have been hypothesized to be the basis of autism phenotypes [7], while neurodevelopmental delay or immaturity is thought to be one of the essential pathophysiologies underlying NDDs [8]. In our prior study, we identified five classes of early neurodevelopmental trajectories, categorized as high normal (11%), normal (49%), low normal (22%), delayed (14%), and markedly delayed (4%), as measured using the Mullen Scale of Early Learning (MSEL) [9] in a sample of Japanese infants followed up from the age of 0 to 2 years [10]. We further showed that the membership of the markedly delayed class, characterized by an overall downward departure from the trajectory of the normal class throughout the two years of follow-up, was predicted by male sex, small for gestational age, low placenta-to-birth weight ratio, and low maternal education. The characteristics of the delayed class, namely, a gradual downward deviation from the normal trajectory that specifically began around the age of 12 months, are more likely to be acquired by a child if he is a male, was born preterm, and had a father who was older. Moreover, the probability of a diagnosis of ASD in the markedly delayed class was highest (32.6%) when compared with the other classes [11]. The downward drift in trajectories can be partly explained by the genetic risk of ASD [12], although the serological basis of class membership in neurodevelopmental trajectories remains to be elucidated.

Immunological factors may be involved in the pathophysiology of NDDs [13], including alterations in the prenatal immune environment, which can contribute to the risk of developing ASD [14–17] and other NDDs [18–20]. Similarly, one study in the general population showed a relationship between cytokine levels in the umbilical cord blood and intellectual development in children [21]. Immune activation within the maternal compartment is likely to influence fetal brain development. However, whether these proteins cross the placenta and exert direct actions on the fetal brain remains unknown [19].

Because cytokines play a dual role in the in utero immune response [22, 23] and brain function [24], they are thought to mediate the complex interplay between prenatal immune processes and critical processes of fetal neurodevelopment [25–31], thereby possibly influencing neurodevelopmental trajectories.

Based on this existing knowledge, we hypothesized that abnormalities in the umbilical cord blood concentrations of circulating inflammatory cytokines may be related to neurodevelopmental trajectories in infants. To test this, we investigated the associations between cytokine concentrations in the umbilical cord serum and distinct classes of five neurodevelopmental trajectories during the first two years of life that were already reported in an investigation of a representative birth cohort in Japan.

## Methods and materials

### Study design

This study was conducted as part of an ongoing cohort study, the Hamamatsu Birth Cohort Study for Mothers and Children (HBC Study), the details of which have been described elsewhere [32, 33]. This study was conducted in accordance with the Strengthening the Reporting of Observational Studies in Epidemiology guidelines.

### Ethical issues

The study protocol was approved by the Hamamatsu University School of Medicine and University Hospital Ethics Committee (Ref No. 18–166). Written informed consent was obtained from each caregiver (the mother in most families) for their and their infant's participation. The "Ethical Guidelines for Life Sciences and Medical Research Involving Human Subjects" issued by the Ministry of Education, Culture, Sports, Science and Technology in Japan requires that the written consent of a surrogate be given priority in the participation of children and adolescents under 16 years old in any research. However, efforts to explain and obtain consent were required even for children and adolescents outside this age range (i.e. <16 years), where written consent was not required, and we took action accordingly.

### Participants

Participant recruitment began on November 19, 2007, and ended on March 31, 2012. A consecutive series of 1065 mothers and 1152 infants born between December 24, 2007, and June 30, 2011, were enrolled. All pregnant women in their first or second trimester who underwent a check-up at Hamamatsu University Hospital or Kato Maternity Clinic were eligible to participate in this study. In Japan, pregnant women can select their maternity clinic from a variety of choices, ranging from private clinics to large general hospitals. The demographic characteristics of the participants enrolled in the two institutions were similar, with the exception of the mothers' age. Mothers who came to the Kato Maternity Clinic first were younger than those who visited the Hamamatsu University Hospital first (t = -2.93, P = 0.03). All mothers provided consent to participate in this study, including those who chose the Kato Maternity Clinic for their first visit, but eventually gave birth at Hamamatsu University Hospital. The postnatal assessments were performed in the laboratory. We included all infants in the assessments, regardless of birth weight, week of gestation, or Apgar score. Based on reports from the Department of Health, Labor, and Welfare, Japan, we determined that the participating mothers in the study were representative of the demographic characteristics of typical Japanese mothers in terms of age, socioeconomic status, and parity. Similarly, their infants represented a typical population of children with regard to birth weight and gestational age at birth. Therefore, it can be argued that the participating cohort represents the general Japanese population.

### Neurodevelopmental trajectory classes

This study adapted our previous neurodevelopmental classification that utilized latent class growth analyses of MSEL measurement data from participants in an HBC study [10].

The MSEL is a composite scale comprising five subscales that assess gross motor function, visual reception, fine motor function, and receptive and expressive language [9]. Measurements were taken when the infants reached 1, 4, 6, 10, 14, 18, and 24 months of age. We excluded 183 participating mothers and 198 infants who missed five or more of the total seven follow-up evaluations after birth. We further excluded two mother–infant dyads as the infants

were diagnosed with Down syndrome. Thus, 952 infants (82.6%) and 880 mothers (82.6%) were included in the analyses.

Latent class growth analysis was applied to distinguish between distinct subgroups of individuals, termed latent classes, following distinct patterns of change over time. To investigate neurodevelopmental trajectories, we applied a parallel process latent class growth analysis which allowed simultaneous processing of five-domain data and contained combinations of continuous latent growth variables including the intercept, slope, the quadratic term for each of the five sub-domains of outcome and a latent categorical variable. To examine whether there was any association between missingness and neurodevelopmental growth patterns, we examined the interaction of group (excluded vs. included) × slope using a linear mixed model. There were no differences in the MSEL scores between the excluded infants and those included in the analysis.

Five classes of neurodevelopmental trajectories were identified: high normal (N = 110, 11.5%), normal (N = 468, 49.2%), low normal (N = 202, 21.2%), delayed (N = 134, 14.1%), and markedly delayed (N = 38, 4.0%). An overall delay in the early developmental stages was characterized by a markedly delayed class. Notably, such a delay first became salient in gross and fine motor domains and later in language domains, especially receptive language. The delayed class showed trajectories with the same convex curve and downward turn with ageing in all five domains. The low normal class showed a slight delay in the early developmental stages, but had caught up by 24 months. The markedly delayed class could be predicted by male sex, small for gestational age, low placenta-to-birthweight ratio, and low maternal education. The delayed class could be predicted by male sex, preterm birth and advanced paternal age. The results remained identical when multiple imputation was applied.

## Cytokine measurements

Blood samples were obtained from the clamped umbilical vein immediately after and before delivery of the placenta. The samples were collected using a Vacutainer blood collection system and centrifuged. Supernatants were stored in polypropylene tubes at -80˚C until testing. Serum concentrations of interleulin (IL)-1α, IL-1b, IL-2, IL-4, IL-5, IL-6, IL-8, IL-10, IL-12p70, IL-13, IL-15, IL-17, IL-23, interferon (IFN)-γ, tumor necrosis factor (TNF)-α, TNF-β, eotaxin, C-C motif chemokine ligand (CCL)-2, CCL-3, growth-regulated oncogene (GRO)α, I-309, interferon-inducible protein (IP)-10, regulated on activation, normal T cell expressed and secreted (RANTES), thymus and activation-regulated chemokine (TARC), angiopoietin (Ang)-2, fibroblast growth factor (FGF), hepatocyte growth factor (HGF), platelet-derived growth factor (PDGF), tissue inhibitor of metalloproteinase (TIMP)-1, TIMP2, and vascular endothelial growth factor (VEGF) were determined using a Q-Plex Human Customs Array multiplexed ELISA (Quansys Biosciences, UT.).

Among the 952 participants in the latent class analysis, 656 samples (high normal (N = 85, 13.0%), normal (N = 322, 49.1%), low normal (N = 137, 20.9%), delayed (N = 87, 13.3%), and markedly delayed (N = 25, 3.8%) were selected for the measurements based on minimal hemolysis on visual recognition.

## Statistical analysis

Among the 31 cytokines measured, six (IL-4, IL-6, IL-15, IL-17, IFN-γ, and FGF) were excluded from the statistical analysis because fewer than 50% of the samples were successfully measured, leaving 25 cytokines for the statistical analysis.

Outliers, defined as values outside the limits of 3.09 standard deviations (cumulative probability value of 99.9%) [34], were excluded from the analysis. Cytokine concentrations were ln-

transformed to normalize their distribution and treated as a standardized score (z-score). Missing values were handled using multiple imputations with chained equations (MICE), assuming that they were randomly missing. The missing values were imputed to create 20 complete datasets using Rubin's rules [35], and the results were integrated to obtain a summarized estimation. All of the variables investigated in the analysis were included in the imputation model.

Multinomial logistic regression was used to estimate the association between each cytokine level and the five neurodevelopment classes, with the normal class as the base outcome (single-cytokine analyses). Given the exploratory nature of the analyses, the significance level was set at $P < 0.05$. Next, we conducted a multi-cytokine analysis, including cytokines showing a significant association in the single cytokine analyses. Referring to our previous study [10], we further adjusted for sex, birth weight, gestational age, paternal age, and history of maternal education as covariates using a multi-cytokine analysis. In addition, we adjusted for cesarean section, considering the difference in cord blood cytokine levels after vaginal delivery [36, 37]. Multi-cytokine analysis was performed for multiple comparisons of the included cytokines. All analyses were performed using STATA ver.14.0.

## Results

### Participants

The descriptive statistics of the study population are displayed in Table 1.

The descriptive statistics of the 656 participants included in the analyses were not significantly different from those of the nonparticipants utilized in the latent class growth analyses.

**Table 1. Characteristics of participating infants and their parents.**

|  | N = 656 |
| --- | --- |
|  | **Mean (SD)** |
| Birthweight (g) | 2981.5 (415.2) |
| Gestational age at birth (weeks) | 39.1 (1.4) |
| Paternal age at birth (years) | 33.3 (5.7) |
| Maternal age at birth (years) | 31.6 (5.0) |
| Paternal education (years) | 14.1 (2.6) |
| Maternal education (years) | 13.9 (1.9) |
|  | N (%) |
| Sex |  |
| Male | 341 (52.0) |
| Female | 315 (48.0) |
| Cesarean section |  |
| Vaginal | 504 (76.8) |
| Elective | 105 (16.0) |
| Emergency | 47 (7.2) |
| Trajectory class |  |
| High normal | 85 (13.0) |
| Normal | 322 (49.1) |
| Low normal | 137 (20.9) |
| Delayed | 87 (13.3) |
| Markedly delayed | 25 (3.8) |

Abbreviation: SD, standard deviation

## Cytokine measurements

As shown in Table 2, IL-4, IL-6, IL-15, IL-17, IFNγ-, and FGF were detected in < 50% of the samples, and were therefore excluded from subsequent analyses.

## Multinomial logistic regression

The results of the single- and multi-cytokine analyses are shown in Tables 3 and 4, respectively.

In the single-cytokine analyses, IL-23, CCL-2, TARC, and HGF were found to be significantly associated with specific class membership in the neurodevelopmental trajectories. In the

**Table 2. Cytokine concentrations in cord blood.**

| Cytokine | N | % | Mean (pg/ml) | SD (pg/ml) | Minimum (pg/ml) | Maximum (pg/ml) |
|---|---|---|---|---|---|---|
| IL-1α | 465 | 70.9 | 4.12 | 1.48 | 1.33 | 8.05 |
| IL-1β | 517 | 78.8 | 8.6 | 2.42 | 4.45 | 15.84 |
| IL-2 | 564 | 86.0 | 6.01 | 1.18 | 2.9 | 9.62 |
| IL-4 | 172 | 26.2 | 2.3 | 0.61 | 2.18 | 5.57 |
| IL-5 | 561 | 85.5 | 2.33 | 0.44 | 1.16 | 3.78 |
| IL-6 | 101 | 15.4 | 40.22 | 145.53 | 4.85 | 1240.67 |
| IL-8 | 654 | 99.7 | 11.82 | 12.33 | 2.02 | 119.5 |
| IL-10 | 368 | 56.1 | 6.77 | 5.68 | 3.64 | 42.95 |
| IL-12p70 | 388 | 59.1 | 3.34 | 0.65 | 2.41 | 5.57 |
| IL-13 | 476 | 72.6 | 2.83 | 0.86 | 1.5 | 5.48 |
| IL-15 | 201 | 30.6 | 5.09 | 1.26 | 3.52 | 10.59 |
| IL-17 | 23 | 3.5 | 3.25 | 0.24 | 3 | 3.83 |
| IL-23 | 591 | 90.1 | 24.87 | 4.12 | 10.75 | 36.65 |
| IFN-γ | 146 | 22.3 | 11.96 | 5.66 | 8.01 | 65.33 |
| TNF-α | 650 | 99.1 | 11.19 | 3.82 | 1.17 | 23.09 |
| TNF-β | 369 | 56.3 | 5.08 | 1.25 | 3.15 | 9.14 |
| Eotaxin | 650 | 99.1 | 51.24 | 19.62 | 1.91 | 116.25 |
| CCL-2 | 651 | 99.2 | 158.6 | 60.86 | 16.03 | 590.26 |
| CCL-3 | 652 | 99.4 | 19.12 | 6.51 | 5.07 | 39.79 |
| GRO-α | 653 | 99.5 | 18.26 | 11.78 | 1.79 | 77.27 |
| I-309 | 617 | 94.1 | 19.43 | 7.69 | 4.92 | 47.44 |
| IP-10 | 617 | 94.1 | 33.81 | 17.82 | 2.84 | 127.44 |
| RANTES | 652 | 99.4 | 38578.93 | 15641.92 | 93.66 | 86686.23 |
| TARC | 647 | 98.6 | 581.05 | 287.98 | 14.53 | 1517.07 |
| Ang-2 | 462 | 70.4 | 63.91 | 54.14 | 11.68 | 422.07 |
| FGF | 73 | 11.1 | 30.55 | 9.49 | 19.59 | 71.61 |
| HGF | 652 | 99.4 | 1268.79 | 399.94 | 32.49 | 2673.04 |
| PDGF | 653 | 99.5 | 829.64 | 374.82 | 2.64 | 2229.68 |
| TIMP-1 | 656 | 100.0 | 150899.8 | 49080.64 | 3153.47 | 281003.8 |
| TIMP-2 | 656 | 100.0 | 67447.65 | 37586.9 | 1638.7 | 167090 |
| VEGF | 649 | 98.9 | 176.06 | 80.83 | 4.43 | 423.24 |

Abbreviations: Ang-2, angiopoietin-2; CCL, C-C motif chemokine ligand; FGF, fibroblast growth factor; GRO-α, growth-regulated oncogene; HGF, hepatocyte growth factor; IFN, interferon; IL, interleukin; IP, interferon-inducible protein; PDGF, platelet-derived growth factor; RANTES, regulated on activation, normal T cell expressed and secreted; SD, standard deviation; TARC, thymus and activation-regulated chemokine; TIMP, tissue inhibitor of metalloproteinase; TNF, tumor necrosis factor; VEGF, vascular endothelial growth factor

**Table 3. Association of cord serum cytokine levels with the five neurodevelopmental trajectories, as assessed by single-cytokine analyses.**

| Cytokine | | | Trajectory classes | | |
|---|---|---|---|---|---|
| | High normal | Normal | Low normal | Delayed | Markedly delayed |
| | (N = 85, 13.0%) | (N = 322, 49.1%) | (N = 137, 20.9%) | (N = 87, 13.3%) | (N = 25, 3.8%) |
| | OR (95%CI) | Base outcome | OR (95%CI) | OR (95%CI) | OR (95%CI) |
| IL-1α | 1.03(0.80–1.33) | | 0.89(0.71–1.13) | 1.00(0.75–1.33) | 0.61(0.33–1.12) |
| IL-1β | 0.92(0.70–1.20) | | 0.93(0.75–1.16) | 0.91(0.69–1.19) | 0.70(0.44–1.10) |
| IL-2 | 0.99(0.78–1.26) | | 0.97(0.78–1.19) | 0.87(0.67–1.12) | 0.62(0.38–1.01) |
| IL-5 | 0.98(0.76–1.26) | | 0.89(0.72–1.09) | 1.03(0.81–1.32) | 0.87(0.54–1.39) |
| IL-8 | 1.18(0.94–1.47) | | 1.13(0.92–1.40) | 1.17(0.93–1.48) | 1.06(0.78–1.44) |
| IL-10 | 1.20(0.95–1.53) | | 0.86(0.63–1.17) | 0.94(0.66–1.36) | 0.96(0.55–1.67) |
| IL-12p70 | 1.08(0.81–1.43) | | 0.869(0.67–1.12) | 0.95(0.71–1.28) | 0.86(0.48–1.53) |
| IL-13 | 0.97(0.75–1.27) | | 0.97(0.78–1.21) | 0.80(0.59–1.08) | 0.70(0.39–1.28) |
| IL-23 | 1.10(0.85–1.41) | | 0.94(0.77–1.15) | 0.85(0.66–1.09) | 0.50(0.32–0.78)** |
| TNF-α | 0.93(0.73–1.27) | | 1.12(0.92–1.38) | 0.89(0.70–1.14) | 0.77(0.56–1.06) |
| TNF-β | 0.95(0.72–1.35) | | 0.93(0.74–1.17) | 0.95(0.70–1.29) | 0.84(0.50–1.42) |
| Eotaxin | 1.06(0.84–1.35) | | 1.01(0.83–1.40) | 1.11(0.87–1.40) | 1.17(0.73–1.87) |
| GRO-α | 1.06(0.82–1.37) | | 1.14(0.93–1.39) | 1.03(0.83–1.28) | 1.14(0.77–1.68) |
| I-309 | 0.99(0.78–1.26) | | 1.03(0.84–1.39) | 0.86(0.68–1.10) | 0.80(0.52–1.21) |
| IP-10 | 1.06(0.84–1.35) | | 1.06(0.87–1.30) | 0.90(0.71–1.15) | 1.08(0.81–1.44) |
| CCL-2 | 1.07(0.81–1.41) | | 1.24(1.03–1.48)* | 0.93(0.70–1.24) | 0.85(0.52–1.39) |
| CCL-3 | 1.25(0.99–1.57) | | 1.06(0.86–1.30) | 0.94(0.73–1.21) | 1.07(0.67–1.69) |
| RANTES | 0.90(0.70–1.17) | | 0.90(0.73–1.11) | 1.11(0.87–1.41) | 0.72(0.45–1.13) |
| TARC | 1.41(1.12–1.77)** | | 1.12(0.91–1.38) | 0.94(0.73–1.21) | 1.01(0.64–1.60) |
| Ang-2 | 1.09(0.84–1.41) | | 1.01(0.82–1.25) | 1.11(0.87–1.40) | 1.28(0.90–1.85) |
| HGF | 1.19(0.94–1.49) | | 0.83(0.67–1.02) | 0.77(0.59–0.99)* | 0.93(0.61–1.43) |
| PDGF | 1.03(0.80–1.33) | | 1.08(0.89–1.31) | 1.03(0.81–1.31) | 1.09(0.73–1.62) |
| TIMP-1 | 1.01(0.81–1.36) | | 0.96(0.78–1.18) | 0.98(0.78–1.24) | 1.17(0.82–1.67) |
| TIMP-2 | 0.81(0.6–41.03) | | 1.05(0.85–1.29) | 0.97(0.76–1.24) | 0.73(0.48–1.10) |
| VEGF | 1.20(0.95–1.53) | | 1.04(0.86–1.27) | 0.88(0.69–1.13) | 1.00(0.63–1.60) |

Abbreviations: Ang, angiotensin II; CCL, C-C motif chemokine ligand; Cl, confidence interval; FGF, fibroblast growth factor; GRO, growth-regulated oncogene; HGF, hepatocyte growth factor; IFN, interferon; IL, interleukin; IP, interferon-inducible protein; OR, odds ratio; PDGF, platelet-derived growth factor; RANTES, regulated on activation, normal T cell expressed and secreted; TARC, thymus and activation-regulated chemokine; TIMP, tissue inhibitor of metalloproteinase; TNF, tumor necrosis

*P<0.05

**P<0.01

***P<0.001

multi-cytokine analysis (Table 4, Fig 1, and S1 Fig), IL-23 was found to be significantly associated with the markedly delayed class (odds ratio (OR):0.44, 95%confidence interval (CI):0.26–0.73), indicating that lower levels of IL-23 in the cord serum is associated with an increased likelihood of assignment to the markedly delayed class. Further, TARC was significantly associated with the class membership of the high normal class (OR:1.37, 95%CI:1.06–1.75) and low normal class (OR:1.28, 95%CI:1.01–1.61), while CCL-2 and HGF were not associated with class membership in the multi-cytokine analyses.

## Sensitivity analysis

To explore the robustness of our findings, we repeated the multi-cytokine analysis without using the imputed values. Overall, the results remain unchanged. IL-23 remained associated

**Table 4. Association of cord serum cytokine levels with the five neurodevelopmental trajectories, as assessed by multi-cytokine analyses.**

| Cytokine | | | Trajectory classes | | |
|---|---|---|---|---|---|
| | High normal | Normal | Low normal | Delayed | Markedly delayed |
| | (N = 85, 13.0%) | (N = 322, 49.1%) | (N = 137, 20.9%) | (N = 87, 13.3%) | (N = 25, 3.8%) |
| | OR (95%CI) | Base outcome | OR (95%CI) | OR (95%CI) | OR (95%CI) |
| IL-23 | 1.10(0.84–1.43) | | 0.92(0.76–1.13) | 0.86(0.66–1.12) | 0.44(0.26–0.73)*** |
| CCL-2 | 1.08(0.81–1.43) | | 1.09(0.91–1.31) | 0.84(0.62–1.15) | 0.78(0.51–1.21) |
| TARC | 1.37(1.06–1.75)* | | 1.28(1.01–1.61)* | 1.05(0.79–1.38) | 1.04(0.64–1.70) |
| HGF | 1.03(0.78–1.35) | | 0.88(0.70–1.12) | 0.84(0.65–1.09) | 0.98(0.62–1.54) |

Abbreviations: CCL, C-C motif chemokine ligand; CI, confidence interval; HGF, hepatocyte growth factor; IL, interleukin; OR, odds ratio; TARC, thymus and activation-regulated chemokine.

$*P < 0.05$

$**P < 0.01$

$***P < 0.001$

with the markedly delayed class (OR:0.35, 95%CI:0.35–0.45, $P < 0.001$). TARC remained associated with the high normal class (OR:1.42, 95%CI:1.34–1.51, $P < 0.001$) and low normal class (OR:1.23, 95%CI:1.16–1.30, $P < 0.001$).

## Discussion

This study investigated whether cytokine profiles at birth could predict early neurodevelopmental trajectories. Overall, the results showed that decreased IL-23 levels in cord blood were associated with membership assignment to the markedly delayed class, even after adjusting for potential confounders. Furthermore, IL-23 levels decreased as the developmental trajectory was delayed, indicating that IL-23 concentration play an important role in development and could be a useful marker to predict developmental trajectory at birth.

IL-23 is a pro-inflammatory cytokine produced by activated dendritic cells and macrophages in response to microbial pathogens. IL-23 is key in Th17 cell development and IL-17 expression [38, 39]. IL-17 is further implicated in the pathogenesis of several inflammatory disorders, including neurological diseases. Consequently, the IL-17 pathway is a key drug target in many autoimmune and chronic inflammatory disorders, and therapeutic monoclonal antibodies targeting the IL-23-IL-17 pathway have been found to be highly effective in some of these diseases [40].

Several past human studies have demonstrated decreased concentrations of IL-23 in plasma and lower production of IL-23 following immune activation (phytohemagglutinin activation) of peripheral blood mononuclear cells in children with ASD at age 2–5 years, and these studies also showed that IL-17 concentration in plasma and IL-17 production following immune activation of peripheral blood mononuclear cells were not different from typically developing children [41, 42]. Furthermore, IL-23 is reportedly associated with ASD [43]. More recently, a large genetic study conducted using PGC and iPSYCH showed that genetic variants linked to the expression of IL-23 were associated with autism [44]. Although the precise mechanism underlying the association between IL-23 and neurodevelopment has not been clarified, in line with our findings, human studies have nevertheless indicated the possible protective roles of IL-23 in neurodevelopment (for review, see Hughes et al., 2023) [45].

The innate immune system comprises three components, involving sensor, effector, and regulatory functions. When sensors are activated in the innate immune system, downstream effectors and the regulatory function are initiated [46, 47]. Pattern recognition receptors are a

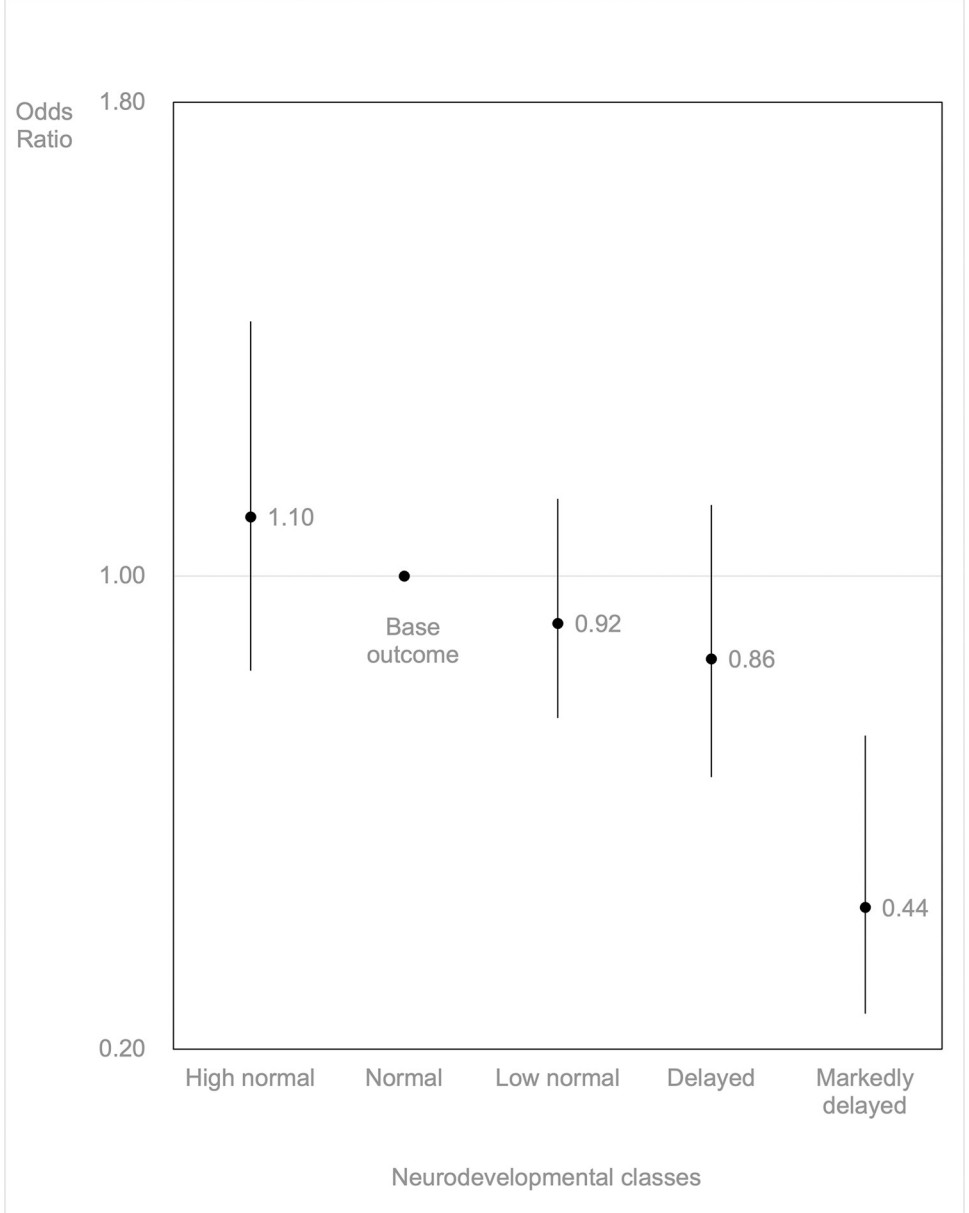

**Fig 1. Odds ratio and 95% confidence interval of the association between Interleukin—23 and neurodevelopmental class.** The dots indicate Odds ratio, and the bars indicate 95% confidence interval.

type of innate sensors that include Toll-like receptors (TLRs), which have attracted significant attention from researchers in recent years [48]. Recently, TLR expression in the syncytiotrophoblast, which covers the placental villi and comes into direct contact with pathogens in the maternal blood, was shown to play a key role in placental host defense [49]. The pattern of changes in cytokine production throughout development from premature infants to adults after TLR stimulation has been thoroughly described in a previous review [48]. When cytokines are released by TLR stimulation, they can exert significant regulatory effects on innate and adaptive immune cells. For example, after TLR stimulation of whole blood, anti-inflammatory innate cytokines (IL-10) are produced in the largest amount in preterm infants, while

the innate immune system of term infants predominantly produces Th17 cell-promoting cytokines, such as IL-6 and IL-23 [50–52]. The vast majority of cord blood samples used in this study were extracted from term infants, and the statistical analyses were adjusted for gestational age. Therefore, low concentrations of IL-23 in the cord blood serum of the markedly delayed class may reflect the abnormal innate TLR stimulation mechanisms, consequently determining neurodevelopmental trajectories. Phytohemagglutinin is a human TLR agonist [53]. Therefore, the lower production of IL-23 following phytohemagglutinin activation in children with ASD [42] suggests the existence of an abnormal TLR stimulation mechanism.

Increased TARC levels in cord blood was associated with high and low-normal classes. TARC is a CC chemokine commonly associated with type 2 immune responses, and several studies have shown that TARC levels may increase in the serum, tissues, or both, under varying eosinophilic conditions [54]. The results concerning TARC were difficult to interpret, as the associations showed opposite directions compared with IL-23, and the levels of TARC did not affect the delayed classes.

A previous study reported that reduced IFN-γ and IL-12p70 levels in the umbilical cord serum were associated with low-performance IQ (<70), indicating a possible protective role of IFN-γ and IL-12p70 during neurological development [21]. However, IL-23 and TARC levels were not measured in this prior study, and our current study excluded IFN-γ from the analyses because of the shortage of measured samples. Furthermore, according to the primary purpose of investigating small-for-gestational-age children, the study population consisted mainly of low-income children. A recent study showed a role for maternal immune activity in fetal neurodevelopment, which is partly exacerbated by socioeconomic disadvantages [55]. Therefore, the difference between the findings of our and previous studies [21] (i.e., the absence of an association between cord blood IL-12p70 levels and neurodevelopment) could be explained by differences in socioeconomic status.

Infection during pregnancy activates the maternal immune system to enhance the production of proinflammatory cytokines, such as IL-6, IL-17, and TNF-α. These cytokines can cross the placenta and enter the fetal circulation, causing abnormalities in brain development and behavior [56]. Because we did not measure IL-6 and IL-17 levels, a general discussion of the effects of maternal immune activation via these two cytokines are difficult. As for TNF-α, one study found that cord blood TNF-α levels were associated with a specific neurodevelopmental domain (prosocial behavior), but was not associated with other domains (hyperactivity / inattention) [57]. These results imply that not all domains of brain function were damaged by deviations in cord blood TNF-α levels, which partly explains the difference from our current results (i.e., absence of association between cord blood TNF-α levels and neurodevelopment).

This study has several limitations which should be mentioned. Firstly, several cytokines, including IL-6, IL-17, and IFN-γ, were not analyzed, mainly due to technical difficulties in measurement. The lack of data on IL-23-related cytokines limits the interpretation of the association between IL-23 levels and neurodevelopment. Therefore, further studies that improve the measurement technique are needed to demonstrate the precise nature of the effects of IL-23 and related cytokines on neurodevelopment. Second, we could not measure the actual central nervous system concentrations of cytokines in the CNS. However, evidence suggests that cytokines pass through and exert activity beyond the blood-brain barrier in utero [29].

The primary strength of this study is that we examined the association between prenatal cytokinesis and neurodevelopment using detailed repeated assessments of infant developmental trajectories in a sample reflecting standard Japanese population characteristics. Consequently, we identified a critical role of IL-23 in neurodevelopment during infancy. Confounding is unlikely to explain the results reported in this study, as the final models controlled for all important confounders in this sample. Moreover, we can rule out observer and

reporting bias, as the cytokine analysis was conducted with the investigators blinded to the neurodevelopmental outcome classes.

In conclusion, we found that low cord blood IL-23 levels predict delayed early neurodevelopment. Further epidemiological research closely integrating in utero molecular markers is required to advance our understanding of the complex mechanisms underlying early human neurological development. Nevertheless, the results of this study may help to identify predictive markers of potentially susceptible subgroups at birth and ultimately move beyond descriptive syndromes in NDDs, towards a nosology informed by biological bases.

## Supporting information

**S1 Fig. Cord blood concentrations of Interleukin-23 in participants stratified by neurodevelopmental classification.** Interleukin-23 concentrations are shown as box-and-whisker plots; midlines indicate medians, boxes indicate interquartile values, whiskers indicate upper and lower adjacent values (the values in the data that are farthest away from the median on either side of the box but are still within a distance of 1.5 times the interquartile range from the nearest end of the box), and dots indicate outside values.
(TIF)

**S1 File. Minimal underlying data set.**
(XLS)

## Acknowledgments

The authors would like to thank Noriyoshi Takei for performing the early planning of the HBC study.

## Author Contributions

**Conceptualization:** Tomoko Nishimura, Hitoshi Kuwabara, Kenji J. Tsuchiya.

**Data curation:** Tomoko Nishimura, Hitoshi Kuwabara, Kenji J. Tsuchiya.

**Formal analysis:** Tomoko Nishimura, Hitoshi Kuwabara.

**Funding acquisition:** Kenji J. Tsuchiya.

**Investigation:** Machiko K. Asaka, Tomoko Nishimura, Hitoshi Kuwabara, Kenji J. Tsuchiya.

**Methodology:** Hitoshi Kuwabara.

**Project administration:** Machiko K. Asaka, Tomoko Nishimura, Hitoshi Kuwabara, Hiroaki Itoh, Kenji J. Tsuchiya.

**Resources:** Hiroaki Itoh, Kenji J. Tsuchiya.

**Software:** Tomoko Nishimura.

**Supervision:** Tomoko Nishimura, Hitoshi Kuwabara, Nagahide Takahashi, Kenji J. Tsuchiya.

**Validation:** Tomoko Nishimura, Hitoshi Kuwabara, Kenji J. Tsuchiya.

**Writing – original draft:** Machiko K. Asaka, Tomoko Nishimura, Hitoshi Kuwabara.

**Writing – review & editing:** Machiko K. Asaka, Hitoshi Kuwabara, Nagahide Takahashi, Kenji J. Tsuchiya.

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
