## [Decision Letter · Decision Letter 0]

29 Jan 2024

PONE-D-23-33306IL-23 levels in umbilical cord blood associated with neurodevelopmental trajectories in infancyPLOS ONE

Dear Dr. Kuwabara,

Thank you for submitting your manuscript to PLOS ONE. After careful consideration, we feel that it has merit but does not fully meet PLOS ONE’s publication criteria as it currently stands. Therefore, we invite you to submit a revised version of the manuscript that addresses the points raised during the review process.

We look forward to receiving your revised manuscript.

Kind regards,

Wen-Jun Tu

Academic Editor

PLOS ONE

Reviewers' comments:

Reviewer's Responses to Questions

**Comments to the Author**

1. Is the manuscript technically sound, and do the data support the conclusions?

Reviewer #1: No

Reviewer #2: Yes

Reviewer #3: Yes

Reviewer #4: Partly

2. Has the statistical analysis been performed appropriately and rigorously? 

Reviewer #1: Yes

Reviewer #2: Yes

Reviewer #3: Yes

Reviewer #4: No

3. Have the authors made all data underlying the findings in their manuscript fully available?

Reviewer #1: Yes

Reviewer #2: Yes

Reviewer #3: Yes

Reviewer #4: Yes

4. Is the manuscript presented in an intelligible fashion and written in standard English?

Reviewer #1: No

Reviewer #2: Yes

Reviewer #3: Yes

Reviewer #4: Yes

5. Review Comments to the Author

Reviewer #1: Comments.

*Title

- ***** All abbreviations should have been provided in full on first mention and this applies to the title, running (short) title, abstract, impact statement, main text and each table / figure independently as they will be read independently. Please use the abbrevations correctly and effectively so all the abbrevations should be checked.

*Abstract

- Please give the number of the participants for each group.

- The abstract must contain enough data (figures) to enable the reader (researcher) to judge its content and therefore relevance. It can not include undefined terms or contradictory statements.

- The manuscript should have ended with a clear conclusion that sums up the value of the study within its limitations.

*Keywords

- Please add the keywords according to alpbabetical order.

* Introduction

- An introduction should include information on why the subject under investigation is important, what is currently known on the topic and how the current investigation will improve our knowledge.

*Materials and Methods

*Results

*Tables and Figures

* Discussion

- In the discussion the authors should summarise and qualify their findings, compare outcomes with other publications and draw conclusion. The implications of the study and the future direction of study should be discussed. Importantly the study’s limitations should be considered.

- The manuscript should have ended with a clear conclusion that sums up the value of the study within its limitations.

*References

- The number of the references should not be exceeded 30 (n<30) and also should be cited in the last five years.

***** All abbreviations should have been provided in full on first mention and this applies to the title, running (short) title, abstract, impact statement, main text and each table / figure independently as they will be read independently. Please use the abbrevations correctly and effectively so all the abbrevations should be checked.

***** Passive sentences should be prefferred instead of active sentences starting with we.

***** Language and presentation require improvement.

***** Need several grammar/wording corrections (There are a lot of words with no space between them, there are missing spaces after commasin the manuscript and also, there are a lot of gramatical error. Language could be improved).

Reviewer #2: Thank you very much to the authors for presenting this interesting study on the association between cord blood cytokine levels and neurodevelopmental outcome.

The research objectives, methodology and results are clearly presented.

My questions and comments to the authors are as follows:

1) Baseline demographics:

Did the authors compare baseline demographics between the groups to see if they are comparable? Although all the babies are delivered full term, other factors including birth weight, antenatal problems, Apgar scores, chorioamnionitis, neonatal course, family history of developmental delay etc…. may all affect neurodevelopmental outcome. Can the authors elaborate more on this?

2) Neurodevelopmental outcomes:

The authors have used the Mullen Scale of Early Learning to assess neurodevelopmental outcome in the first two years of life. For infants who are assessed to be in the “markedly delayed” or “delayed” group, would there be interventions or further investigations? And are there ongoing studies to look at neurodevelopmental outcome at a later age (eg 5 years old and later)?

3) Cytokine results:

In previous studies on neurodevelopment of neonates (both preterm and term), cytokines including IL-6, IL-17, TNF- γ and TNF-a are associated with inflammation and injury to the developing brain, leading to poor neurodevelopmental outcome. In this study, only IL-23 is found to have significant association. The authors have postulated that the difference in TNF- γ may be due to difference in socioeconomic factors between subjects in various studies. Can the authors explain the difference between these results and previous findings?

Reviewer #3: Dear Authors,

i would like to congratulate You to describe such interesting subject

I would be satisfied if You could desribe in the Introduction IL23 referring to development.

In the line 113 You wrote when child was assessed: could You explain or correct why you wrote: the ages of 1, 24, 6, 10, 14, 18, or 24 months . 24 month is after 1st month, I do not understand?

Could You write more about classification of The Mullen Scale of Early Learning (MSEL) due to next authors or readers have clear information when a child is :"low normal" delayed? Could You write more about exclussion criteria? Were there children with any inflamantory process because of sepsa or sth? Were there any USG of Transparietal ultrasound? Could You write about birth weight, week of gestation or Apgar score in inclussion criteria (VLBW, ELBW, Apgar score under 7 points)? Could You write in disccusion why You suspect ( in which process) children could have abnormal IL23 level?

Your sincerely,

Reviewer #4: This study, using data from the Mullen Scale of Early Learning on over 1000 Japanese infants followed from birth to 2 years, identified five distinct classes of early neurodevelopmental trajectories: high normal, normal, low normal, delayed, and markedly delayed. This classification was proved to be able to associate the cord blood cytokine concentrations with neurodevelopmental outcomes. However, some methodological issues required a revised analysis for robust and conclusive findings.

1) The paper did not mention how the neurodevelopmental trajectories were classified. as the classification of the 5 classes is the key in this paper, detailed explanation of the criteria and methods used to classify infants into the five neurodevelopmental trajectories is necessary. Also please ensure the robustness of the classification techniques.

2) There are about 30 cytokine concentrations considered in the association analyses. However, in the regression analyses p-values were not adjusted for multiple comparison in Table 3. Application of p-value adjustment methods in the regression analysis is necessary, specifically in Table 3.

3) is there a specific reason why the infants' age was not controlled as a confounding factor?

4) It is better to show some figures such as boxplots or histograms for the identified cytokine concentrations among the 5 classes. Generation of boxplots or histograms to visually compare cytokine concentration distributions across the five identified classes will give us more intuitive comparison of the distributions and outliers.

6. PLOS authors have the option to publish the peer review history of their article (what does this mean?). If published, this will include your full peer review and any attached files.

Reviewer #1: **Yes: **Hasan Ali İnal

Reviewer #2: **Yes: **Genevieve Po Gee Fung

Reviewer #3: **Yes: **Roksana Malak Dep of Rheumatology Rehabilitation and Internal Medicine Poland

Reviewer #4: No

---

## [Author Response · Author response to Decision Letter 0]

8 Mar 2024

We would like to thank the Reviewers for their insightful comments and suggestions, which we believe have greatly helped us to improve our manuscript and provide a more balanced account of our research. We have carefully reviewed our manuscript in accordance with these comments, and made all necessary changes, which have shown using the track changes function in the revised manuscript to facilitate the review process. Please find below our point-by-point responses to all of the reviewer comments.

Reviewer #1: 

Title

All abbreviations should have been provided in full on first mention and this applies to the title, running (short) title, abstract, impact statement, main text and each table / figure independently as they will be read independently. Please use the abbreviations correctly and effectively so all the abbreviations should be checked.

Response: We would like to thank you for this valuable suggestion, we have revised the manuscript accordingly. 

Abstract

Please give the number of the participants for each group. The abstract must contain enough data (figures) to enable the reader (researcher) to judge its content and therefore relevance. It cannot include undefined terms or contradictory statements.

Response: We would like to thank you for this valuable suggestion, we have revised the manuscript accordingly. (Lines 1-16)

Keywords

Please add the keywords according to alphabetical order.

Response: Thank you for pointing this out, we have now rearranged the order of keywords. 

An introduction should include information on why the subject under investigation is important, what is currently known on the topic and how the current investigation will improve our knowledge. In the discussion the authors should summarize and qualify their findings, compare outcomes with other publications and draw conclusion. The implications of the study and the future direction of study should be discussed. Importantly the study’s limitations should be considered. The manuscript should have ended with a clear conclusion that sums up the value of the study within its limitations.

Response: We would like to thank you for your comment, and apologize for the lack of clarity on this point we have also referred to the comments of Reviewer#2-4 together, and have made the corresponding modifications to our manuscript. 

References

The number of the references should not be exceeded 30 (n<30) and also should be cited in the last five years.

Response: We would like to thank you for your valuable suggestion; however, PLOS ONE does not limit the number of references, or provide guidelines on how recent the references cited should be.

Passive sentences should be preferred instead of active sentences starting with we. Language and presentation require improvement. Need several grammar/wording corrections (There are a lot of words with no space between them, there are missing spaces after commas in the manuscript and also, there are a lot of grammatical error. Language could be improved).

Response: We would like to thank you for your comment and apologize for this oversight, we have sent the manuscript for English proofreading by a native English speaker again.

Reviewer#2

1) Baseline demographics:

Did the authors compare baseline demographics between the groups to see if they are comparable? Although all the babies are delivered full term, other factors including birth weight, antenatal problems, Apgar scores, chorioamnionitis, neonatal course, family history of developmental delay etc…. may all affect neurodevelopmental outcome. Can the authors elaborate more on this?

Response: We would like to thank you for your valuable suggestion. As you suggested, various confounding factors are likely to influence the results. We selected the baseline factors that showed an association with five neurodevelopmental trajectories, as identified in a previous study by Nishimura et al., 2016, for the adjustments. We have added the description of these associated factors in the method section of the revised manuscript. (Line: 145-147, Line: 186-189)

2) Neurodevelopmental outcomes:

The authors have used the Mullen Scale of Early Learning to assess neurodevelopmental outcome in the first two years of life. For infants who are assessed to be in the “markedly delayed” or “delayed” group, would there be interventions or further investigations? And are there ongoing studies to look at neurodevelopmental outcome at a later age (eg 5 years old and later)?

Response: We would like to thank you for your valuable suggestion. As we did not evaluate the intervention in formulated measure, we were not able to add this data to the current analyses. However, the cohort study (the Hamamatsu Birth Cohort Study for Mothers and Children) is ongoing, and we have published neurodevelopmental transition patterns from infancy to early childhood (Kato et al., Sci Rep. 2022; 12: 4822).

3) Cytokine results:

In previous studies on neurodevelopment of neonates (both preterm and term), cytokines including IL-6, IL-17, INF- γ and TNF-a are associated with inflammation and injury to the developing brain, leading to poor neurodevelopmental outcome. In this study, only IL-23 is found to have significant association. The authors have postulated that the difference in INF- γ may be due to difference in socioeconomic factors between subjects in various studies. Can the authors explain the difference between these results and previous findings?

Response: We would like to thank you for your valuable suggestion. The inconsistency regarding TNF-� might be explained by the difference in outcome measurements between our and previous studies (e.g. Barbosa et al., Brain Behav Immun Health. 2020; 8: 100141). However, as our current experiments did not measure IL-6, IL-17, or INF-γ, it is difficult to interpret the inconsistency between previous studies and our results. We have added these explanations in the revised manuscript. (Line: 316-318, Line: 326-336, Line: 338)

Reviewer #3

I would be satisfied if You could describe in the Introduction IL23 referring to development.

Response: We would like to thank you for your valuable suggestion. As our study was exploratory in nature, and as far as we know, there were no high-quality report that led us to prioritize IL-23 over other cytokines, we thought that it is rather unfair to insist on including IL-23 in the introduction.

In the line 113 You wrote when child was assessed: could You explain or correct why you wrote: the ages of 1, 24, 6, 10, 14, 18, or 24 months. 24 month is after 1st month, I do not understand?

Response: We would like to thank you for your comment and apologize for the error, which we have now corrected. (Line: 122)

Could You write more about classification of The Mullen Scale of Early Learning (MSEL) due to next authors or readers have clear information when a child is: "low normal" delayed? 

Response: We would like to thank you for your valuable suggestion. According to your suggestion, we have added more information regarding the low normal and delayed classes from our previous study by Nishimura et al., 2016. (Line: 142-144) 

Were there children with any inflammatory process because of sepsa or sth? Were there any USG of trans parietal ultrasound?

Response: We would like to thank you for your comment. However, we did not assess sepsa or sth, nor did we measure USG of trans parietal ultrasound.

Could You write more about exclusion criteria? Could You write about birth weight, week of gestation or Apgar score in inclusion criteria (VLBW, ELBW, Apgar score under 7 points)? 

Response: We would like to thank you for your valuable suggestion. According to your suggestion, we have added more information about the inclusion and exclusion criteria, which had been adopted from our previous study by Nishimura et al., 2016. (Line: 107-108, Line: 123-125)

Could You write in discussion why You suspect (in which process) children could have abnormal IL23 level?

Response: We would like to thank you for your valuable suggestion. Although our research design was insufficient to allow us to ascertain the true process of IL-23 affecting the neuro-developments, the review by Kollman et al. (Immunity. 2012; 37(5): 771-83) explained that after Toll-like receptors (TLRs) stimulation of whole blood, production of Th17 cell-promoting cytokines IL-6 and IL-23 dominates in term infant, while further research has shown that low production of IL-23 after phytohemagglutinin activation in children with ASD. As such, we suspected that the abnormality of responses to the stimulation of TLRs in the perinatal period might be involved in the mechanism of abnormal cord blood IL-23 levels. These discussions were already included in the discussion section. (Line: 286-307)

Reviewer #4

1) The paper did not mention how the neurodevelopmental trajectories were classified. as the classification of the 5 classes is the key in this paper, detailed explanation of the criteria and methods used to classify infants into the five neurodevelopmental trajectories is necessary. Also please ensure the robustness of the classification techniques.

Response: We would like to thank you for your valuable suggestion. According to your suggestion, we have added more information about the statistical methods of identification on five neurodevelopmental trajectories from our previous study by Nishimura et al., 2016. (Line: 117-136)

2) There are about 30 cytokine concentrations considered in the association analyses. However, in the regression analyses p-values were not adjusted for multiple comparison in Table 3. Application of p-value adjustment methods in the regression analysis is necessary, specifically in Table 3.

Response: We would like to thank you for your valuable suggestion. Indeed, we performed two step analyses; the first step involved exploratory analyses, while the second (final) step involved multinominal logistic regression, adjusted for multiple comparison. We have added the clear description of adjustments for multiple comparison in revised manuscripts. (Line:189-190)

3) is there a specific reason why the infants' age was not controlled as a confounding factor?

Response: We would like to thank you for your valuable suggestion. At the point of measuring of the Mullen Scale of Early Learning (MSEL) and the sampling of cord blood, all the participants were the same age (month), thus we did not control for age as a confounder. 

4) It is better to show some figures such as boxplots or histograms for the identified cytokine concentrations among the 5 classes. Generation of boxplots or histograms to visually compare cytokine concentration distributions across the five identified classes will give us more intuitive comparison of the distributions and outliers.

Response: We would like to thank you for your valuable suggestion. According to your suggestion, we have generated Fig 1 and S1 Fig. (Line: 232, Line: 248-250, Line: 575-582)

---

## [Decision Letter · Decision Letter 1]

26 Mar 2024

Interleukin-23 levels in umbilical cord blood associated with neurodevelopmental trajectories in infancy

PONE-D-23-33306R1

Dear Dr. Kuwabara,

We’re pleased to inform you that your manuscript has been judged scientifically suitable for publication and will be formally accepted for publication once it meets all outstanding technical requirements.

Kind regards,

Wen-Jun Tu

Academic Editor

PLOS ONE

Additional Editor Comments (optional):

Reviewers' comments:

Reviewer's Responses to Questions

**Comments to the Author**

1. If the authors have adequately addressed your comments raised in a previous round of review and you feel that this manuscript is now acceptable for publication, you may indicate that here to bypass the “Comments to the Author” section, enter your conflict of interest statement in the “Confidential to Editor” section, and submit your "Accept" recommendation.

Reviewer #1: All comments have been addressed

Reviewer #4: All comments have been addressed

2. Is the manuscript technically sound, and do the data support the conclusions?

Reviewer #1: Yes

Reviewer #4: (No Response)

3. Has the statistical analysis been performed appropriately and rigorously? 

Reviewer #1: Yes

Reviewer #4: (No Response)

4. Have the authors made all data underlying the findings in their manuscript fully available?

Reviewer #1: Yes

Reviewer #4: (No Response)

5. Is the manuscript presented in an intelligible fashion and written in standard English?

Reviewer #1: Yes

Reviewer #4: (No Response)

6. Review Comments to the Author

Reviewer #1: Dear Author(s),

Thank you for your revisions. The manuscript was revised accordingly. It can be published.

Reviewer #4: (No Response)

7. PLOS authors have the option to publish the peer review history of their article (what does this mean?). If published, this will include your full peer review and any attached files.

Reviewer #1: **Yes: **Hasan Ali İnal

Reviewer #4: No

---

## [Editor Report · Acceptance letter]

1 Apr 2024

PONE-D-23-33306R1 

PLOS ONE

Dear Dr. Kuwabara, 

I'm pleased to inform you that your manuscript has been deemed suitable for publication in PLOS ONE. Congratulations! Your manuscript is now being handed over to our production team.

Kind regards, 

on behalf of

Dr. Wen-Jun Tu 

Academic Editor

PLOS ONE